# Engineering Antioxidant Surfaces for Titanium-Based Metallic Biomaterials

**DOI:** 10.3390/jfb14070344

**Published:** 2023-06-29

**Authors:** Jithin Vishnu, Praveenkumar Kesavan, Balakrishnan Shankar, Katarzyna Dembińska, Maria Swiontek Brzezinska, Beata Kaczmarek-Szczepańska

**Affiliations:** 1Department of Mechanical Engineering, Amrita Vishwa Vidyapeetham, Amritapuri Campus, Clappana 690525, India; jithinv@am.amrita.edu (J.V.); bala@am.amrita.edu (B.S.); 2Department of Materials Engineering, Indian Institute of Science, Bangalore 560012, India; praveenkk@iisc.ac.in; 3Department of Environmental Microbiology and Biotechnology, Faculty of Biological and Veterinary Sciences, Nicolaus Copernicus University in Toruń, 87-100 Toruń, Poland; 311943@stud.umk.pl (K.D.); swiontek@umk.pl (M.S.B.); 4Department of Biomaterials and Cosmetic Chemistry, Faculty of Chemistry, Nicolaus Copernicus University in Toruń, 87-100 Toruń, Poland

**Keywords:** antioxidant, surfaces, titanium, reactive oxygen species (ROS), biomaterials

## Abstract

Prolonged inflammation induced by orthopedic metallic implants can critically affect the success rates, which can even lead to aseptic loosening and consequent implant failure. In the case of adverse clinical conditions involving osteoporosis, orthopedic trauma and implant corrosion-wear in peri-implant region, the reactive oxygen species (ROS) activity is enhanced which leads to increased oxidative stress. Metallic implant materials (such as titanium and its alloys) can induce increased amount of ROS, thereby critically influencing the healing process. This will consequently affect the bone remodeling process and increase healing time. The current review explores the ROS generation aspects associated with Ti-based metallic biomaterials and the various surface modification strategies developed specifically to improve antioxidant aspects of Ti surfaces. The initial part of this review explores the ROS generation associated with Ti implant materials and the associated ROS metabolism resulting in the formation of superoxide anion, hydroxyl radical and hydrogen peroxide radicals. This is followed by a comprehensive overview of various organic and inorganic coatings/materials for effective antioxidant surfaces and outlook in this research direction. Overall, this review highlights the critical need to consider the aspects of ROS generation as well as oxidative stress while designing an implant material and its effective surface engineering.

## 1. Introduction

One of the key factors associated with inflammatory response is the oxidative stress, which is characterized by the imbalance/disparity between the generation of reactive oxygen species (ROS) and antioxidant defense system [1]. Osteoporosis, the most common bone disorder globally, is a systemic skeletal disorder associated with diminishing bone mass and micro-architectural bone tissue degradation with concomitant bone fragility and osteoporotic fracture [2,3,4]. Considered as one of the major global pandemics of the 21st century, osteoporosis induces more than 8.9 million bone fractures per annum, affecting about 200 million people, and in addition, poses a high risk specifically to post-menopausal women, with 40–50% prevalence in women older than 60 years [5,6,7]. Other leading causes for bone fracture includes road accidents, falls and sports injuries. Following the bone fracture, secondary healing ensues involving various stages such as hematoma formation, acute inflammation, callus formation and bone remodeling [8]. The fracture trauma results in blood vessel rupture in the region of fracture leading to hematoma [9]. The hematoma micro-environment in the fracture site is structurally unstable, hypoxic and acidic, which requires a cross-talk between inflammatory cells and cells related to bone healing in order to re-establish normal homeostatic state [9,10]. The bone remodeling involves a collective involvement of various bone cells such as osteoclasts (removal of damaged and old bones), osteoblasts (synthesis and secretion of osteoid matrix during mineralization) and osteocytes (regulating new bone formation and old bone resorption) [11,12,13]. Oxidative stress is a predominant factor which negatively affects the bone remodeling process resulting in a deteriorated bone mineral density, contributing in the etiology of osteoporosis [14,15,16].

Clinical intervention of bone fractures and defects considers the usage of orthopedic implants for the treatment of orthopedic trauma with minimal harm to the patients. Metallic, ceramic and polymeric biomaterials have been explored and researched for orthopedic implant applications, with each class of materials possessing its own advantages and disadvantages [17,18,19]. Thanks to their superior mechanical properties, metallic materials are the most widely used material for internal fracture fixation components. The three dominant material classes in this aspect are 316L stainless steel, Co-Cr alloys and Ti and its alloys [20,21,22,23]. However, metallic materials are prone to degradation due to corrosion-wear synergy (tribocorrosion) in complex physiological environments capable of eliciting the release of ions and debris in the peri-implant region [24,25]. Such wear-debris release from articulating components can result in the activation and senescence of resident cells including macrophages, fibroblasts, osteoclasts and osteoblasts, eventually leading to the production and release of pro-inflammatory cytokines, chemokines, ROS and reactive nitrogen species (RNS) [26,27]. This elicits chronic inflammatory cascades and oxidative stress reactions eventually resulting in bone resorption and osteolysis induced implant failure [26,28].

During normal healing process, osteoblasts express antioxidant enzymes such as superoxide dismutase (SOD) for inducing the conversion of ROS into O and H_2_O to induce osteoblast differentiation [29,30]. However, during adverse conditions as mentioned above, enhanced ROS activity results in oxidative stress reducing the bone mineralization density by affecting the remodeling process [31,32]. In spite of the presence of endogenous antioxidants, excessive generation of free radicals and inflammatory processes result in oxidative stress [33]. The occurrence of oxidative stress can be ascribed to abnormal activation of enzymes which generates ROS. ROS are highly reactive, short-lived molecules formed as by-products during molecular oxygen reduction which are capable of oxidative damage to macromolecules in biological cells [34,35,36,37,38]. ROS include radical and non-radical oxygen species such as superoxide anion (O^2−^), hydroxyl radical (OH^−^) and hydrogen peroxide (H_2_O_2_) [39,40,41,42]. The mechanism of ROS formation via electrochemical corrosion reaction, radical transformation via Fenton and Haber–Weiss Reactions, light induction and surface catalytic reactions is elaborately reviewed by Kessier et al. [43].

Antioxidants are naturally occurring reducing agents which can hinder the generation of ROS via the phenomenon of scavenging free radicals and eradicating ROS derivatives. Hence, the origin of oxidative stress can be linked to the imbalance between ROS and antioxidants which encompass enzymatic antioxidants (e.g., polyphenols, carotenoids, glutathione, tocopherols) and antioxidant enzymes (SOD, catalase, glutathione peroxidase) [44]. An increment in antioxidant levels can be potentially harmful as it could induce molecular damages, apoptosis or necrosis, and oxidative stress is found to be associated with several diseases including cardiovascular, neurodegenerative, carcinoma, diabetes, ischemia/reperfusion injury, rheumatoid arthritis and aging [45]. Endogenous enzymatic antioxidants include SOD, catalase, glutathione peroxidase and glutathione reductase, whereas non-enzymatic endogenous antioxidants include glutathione and lipoic acid [46].

Metallic implant materials are widely used for bone-anchored therapy for orthopedic and dental treatments. Apart from the wear-induced oxidative stress as discussed above, metallic material insertion during surgical procedure induces large amount of ROS generation and is incapable of generating antioxidants, thereby critically influencing the healing process which elevates the healing time.

Implant surface plays a pivotal role in dictating the host response of the implanted material. In most cases, surface modification of implants alters the surface morphology, topography, chemistry and surface energy, particularly aimed at improving matrix protein adhesion, cellular adhesion and proliferation, to attain better osseointegration [47,48]. A variety of surface modification strategies involving surface texturing and surface coatings have been developed to improve the interfacial mechanical strength, wear resistance, tribo-corrosion resistance and biocompatibility in order to enhance the longevity of orthopedic implants [49,50]. Recently, surface modification of Ti implants has been gaining research attention to repair the impaired osseointegration by developing surfaces with antioxidant activity [51,52,53]. In summary, it is imperative to gain more insights into the advancements in this field to further improve the antioxidant activities of Ti implant surface by proper surface modification to improve its clinical efficiency. In view of these aspects, the present review is focused towards the various surface engineering techniques to combat the undesirable ROS generation associated with Ti-based metallic implants. Several review articles have been published reporting the underlying mechanism of ROS formation and antioxidative mechanisms [30,54,55]. In addition, review articles comprehensively describing surface modification techniques for Ti surface are published [56,57]. The novelty aspect of the present review lies in collating the available reported works in improving the antioxidant properties of Ti-based metallic implant surfaces via various organic and inorganic coatings. Even though several research articles have explored the antioxidant activity of surfaces developed for antibacterial and biocompatible applications, this review exempts these articles and is focused on research associated with surfaces/materials specifically developed for antioxidant purpose. This review initially presents an outline of ROS generation associated with the insertion of Ti implants. This is followed by sections describing organic- and inorganic-based coatings on Ti surfaces to ameliorate the antioxidant aspects along with prospective future perspectives. The major objective of the present review is to provide an overall idea about how surface modification can assist in improving the ROS scavenging activity and reduce oxidative damage to improve the clinical efficiency of Ti-based implants.

## 2. Titanium Alloys and Reactive Oxygen Species Metabolism

Titanium (Ti) and its alloys are the widely used material for a variety of load-bearing orthopedic implant applications thanks to the excellent mechanical aspects, lower modulus values, corrosion resistance and excellent biocompatibility [58]. Ti is a transition metal which exists in a hexagonal closed pack (hcp) crystal structure (α-Ti), which transforms into its allotropic form with a body-centered cubic (bcc) structure (β-Ti) above a temperature of 882 °C, which is retained up to its melting point (1688 °C). Several Ti-based alloys such as commercially pure Ti (cp-Ti, ASTM-F67), Ti-6Al-4V (ASTM-F136), Ti-6Al-7Nb (ASTM-F1472, F1295) and Ti-13Nb-13Zr (ASTM-F1713-08) have been explored for dental implants, bone fixation plates, screws and hip joint stems [59,60]. Current research focus is more shifted towards β-Ti alloys as they possess comparatively lower elastic modulus (as low as 46–55 GPa), high strength, good cold workability and, most importantly, the beneficial biocompatibility aspects due to β-phase stabilizing alloying additions (Nb, Ta, Mo, Mn, Fe etc.) [61,62,63]. In addition, Ti-based shape memory alloys are prospective materials for various biomedical applications owing to the shape memory and super-elasticity effects [64]. Despite these beneficial aspects, wear-induced aseptic loosening is a limiting factor hampering the efficiency of Ti-based orthopedic implants [65]. Wear-particle phagocytosis by macrophages can induce cytokine and free radical release, resulting in an aseptic inflammatory response, capable of promoting osteoclast resorption [66]. The role of high oxidative stress as one of the main causative factors in various inflammatory and degenerative disorders points towards the contribution of ROS towards aseptic loosening. As a response to the released metallic particles in a physiological condition, the immune system elicits an inflammation process, which involves generation of ROS through a series of enzyme-assisted biochemical reactions (schematic figure as shown in Figure 1) [67].

Superoxide radical generation is catalyzed by NADPH (nicotinamide adenine dinucleotide phosphate) oxidase (Equation (1)). Electrons from NADPH is accepted by the cytosolic domain of gp91^phox^ (electron transferase of NADPH oxidase) and is transferred across membrane to O_2_ to generate superoxide radical (O_2_^−^) as the primary product [68]. Gp91^phox^ contains all the required co-factors to effectuate electron transfer reaction, in which electrons transfer from NADPH onto flavin adenine dinucleotide (FAD) and to the haem group in the following step, inducing reduction of O_2_ to O_2_^−^ [69].
(1)$$NADPH+O_{2}↔{NADP}^{+}+O_{2}^{-}+H^{+}$$

In response to this, antioxidant scavenging enzymes such as SOD promote dismutation to convert superoxide to hydrogen peroxide and an oxygen molecule (Equation (2)), which occurs spontaneously (rate constant = 5 × 10^5^ M^−1^s^−1^ at neutral pH) [70]. This reaction is greatly accelerated by SOD, and the corresponding catalytic activity is attributed partly to the electrostatic interactions in active center of SOD protein [71].
(2)$$2O_{2}^{-}+2H^{+}\to O_{2}+H_{2}O_{2}$$

Stimulation of neutrophils results in oxygen consumption in a respiratory burst that produces O_2_^−^ and H_2_O_2_. Simultaneous discharge of abundant myeloperoxidase enzyme occurs, which utilizes H_2_O_2_ to oxidize halides (chlorides, bromides) and thiocyanates to corresponding hypohalous acids and hypothiocyanite [72]. Myeloperoxidase, also called verdoperoxidase, is a heme-containing peroxidase generated mostly from polymorphonuclear neutrophils and found in primary granules of granulocytic cells [73]. The reaction between hydrogen peroxide with halides (such as Cl^−^ in physiological environment) is catalyzed by granule-localized myeloperoxidase to form hypochlorous acid (bleach) (Equation (3)).
(3)$$H_{2}O_{2}+{Cl}^{-}\to HOCl+{OH}^{-}$$

In addition, hydrogen peroxide can generate hydroxide and hydroperoxyl radicals by reacting with ferrous and ferric cations (Fenton reactions). Fenton chemistry can significantly enhance the degradation of many transition metals (including Ti alloys, Co-Cr alloys) [74]. Fenton reaction involves an initial electron transfer with neither bond formation nor breaking and the generation of hydroxyl radicals [75]. Haber Weiss reaction which makes use of Fenton chemistry involves vital mechanism in which highly reactive hydroxyl radical generation occurs [76]. Another possible cathodic reaction taking place at implant/bone interface is oxygen reduction to generate hydrogen peroxide (Equation (4)). The cathodic oxygen reduction can be sub-divided into several reactions, resulting in the generation of hydroxyl radicals and hydrogen peroxide.
(4)$$O_{2}+2H_{2}O+2e^{-}\to H_{2}O_{2}+2{OH}^{-}$$

Hence, ROS are additional products of overall electrochemical reactions occurring in the implant interface other than the metallic ions and/or particles. The presence of ROS (hydroxyl radicals and hydrogen peroxide) can further promote degradation of Ti implants [77]. Among the various ROS molecules, hydrogen peroxide can mix with water and diffuse through membranes of peri-implant tissues, critically affecting intracellular redox status, thereby increasing the chances of implant failure [78].

## 3. Surface Modification for Antioxidant Ti Surfaces

Surface modification of Ti alloys offers an effective strategy to combat the limitations associated with ROS activity. To develop such surfaces/coatings, several surface modification techniques such as layer-by-layer technique, immersion/dip coating, spin coating, plasma immersion ion implantation and radiofrequency plasma-enhanced chemical vapor deposition (enlisted in Table 1) are being researched. A limited number of coating surfaces/materials have been explored to improve the antioxidant activity of Ti surfaces which can be conveniently categorized as organic and inorganic materials for surface modification.

### 3.1. Organic Materials for Surface Modification

#### 3.1.1. Tannic Acid

Tannic acid is a water-soluble natural polyphenol compound, which is often present in tea, wine and fruits and possesses excellent antioxidant and antibacterial activity owing to the presence of pyrogallol and catechol groups [92]. The antioxidant activity of tannic acid is dependent on its capability to chelate metal ions such as Fe(II) and interfering one of the reaction steps in Fenton reaction, thereby retarding oxidation [93]. There are several published review articles pertaining to the surface modification aspects of tannic acid-based coatings for various applications [94,95]. Several techniques such as layer-by-layer deposition [96,97], electrodeposition, UV-assisted deposition [98] and immersion [53] have been used to deposit tannic acid coatings. The presence of catechol groups renders tannic acid substrate-independent adhesive properties. Polyphenol group interactions can occur via several catechol–surface interactions ranging from noncovalent interactions (hydrogen bonding, pi–pi interactions) to chemical bonding (coordination, covalent) [99]. In addition, polyphenol tannic acid is capable of forming functional coatings on various materials by means of an intrinsic auto-oxidative surface polymerization. Sebastian et al. investigated the deposition kinetics of tannic acid on Ti surfaces which revealed a multiphase layer generation [100]. An initial growth phase revealed build-up of layer which is compact as well as rigid (approx. 2 h), followed by adsorption of an increasingly dissipative layer (approx. 5 h). Following this, a coating discontinuation was observed which was corroborated with large particle precipitation in coating solutions.

In order to develop multifunctional coatings on Ti surface, tannic acid is often co-deposited along with other functional biomaterial coatings for prospective implant applications. Hydroxyapatite (Ca_10_(PO_4_)_6_(OH)_2_) is a bioactive material, the main inorganic bone component which possesses excellent osteoinduction and osteoconduction properties. In view of rendering Ti surfaces (which are bioinert) with bioactive and antioxidant properties, hydroxyapatite and tannic acid based composite coatings have been explored. A consistent and strong antioxidant activity was displayed by hydroxyapatite/tannic acid coatings deposited on Ti substrates modified by titania (TiO_2_) nanotubes (Figure 2a–c) [101]. Gelatin added to hydroxyapatite can improve the osteogenic aspects to enhance bone formation. However, gelatin-hydroxyapatite coatings failed in bone conduction function due to weak bonding between them. Tannic acid has been found to strongly adsorb to hydroxyapatite surface and firmly glued gelatin and hydroxyapatite [96]. The resultant tannic acid-hydroxyapatite-gelatin complex coating demonstrated significantly higher antioxidant activity and reduced cell damage/changes in the presence of H_2_O_2_. There are limitations reported with the adherence of tannic acid onto hydroxyapatite and salivary acquired pellicle peptide modified tannic acid exhibited better adsorption performance on hydroxyapatite surface [102]. Tightly adsorbed coating exhibited smooth, superhydrophilic surface with excellent antibacterial and antibiofouling performance.

In order to develop multifunctional antioxidant and antibacterial coatings, tannic acid is coated along with antibacterial elements which can be contact killing, release killing or anti-adhesive. Despite being widely explored for a wide spectrum of antibacterial applications, silver (Ag) usage for bio-surfaces is limited by dose dependent cytotoxicity. Hydroxyapatite-tannic acid coating developed by immersion technique on a Ag-loaded TiO_2_ nanotubular Ti surface demonstrated high antibacterial activity, improved cytocompatibility and revealed slow release of tannic acid from surface, which contributed towards persistent antioxidant activity as shown in Figure 2d–f [53]. Polyethylene glycol is a promising antifouling polymeric interface, an appropriate proton acceptor and can generate hydrogen bonds with tannic acid [103]. Simultaneous deposition of polyethylene glycol resulted in increased coating thickness and improved surface coverage [104]. A novel pH-bacteria triggered antibiotic release mechanism has been developed by layer-by layer deposition of tannic acid with cationic antibiotics such as tobramycin, gentamicin and polymyxin B [105]. Unlike linear polymer molecules which are incapable of retaining antibiotics, tannic acid through its hydrogen bonding and electrostatic interactions interacted well with the antibiotics. The interesting aspect is the non-eluting characteristic of the tannic acid/antibiotic coating which is capable of triggering antibiotic release created by pH reduction induced by bacterial pathogens. Hizal et al. reported an ultrathin tannic acid/gentamicin layer-by-layer film on 3D nano-pillared structures, which exhibited a 10-fold decrease in bacterial attachment due to larger surface area of nanostructured surface and lower bacterial adhesion forces on nanopillar tips [106]. Apart from these, strontium (Sr^2+^) incorporated tannic acid functionalized on Ti surface revealed enhanced osteoblast differentiation and reduced osteoclast activity [107].

#### 3.1.2. Chitosan

Chitosan is a polycationic natural macromolecule (with a molecular structure of (1,4)-linked 2-amino-2-deoxy-β-d-glucan), which is capable of reacting with many physiologically relevant ROS [108,109,110,111]. Owing to its various beneficial aspects such as improvement of osseointegration and cellular interactions, minimal foreign body response, favorable degradation rate and, most importantly, due to antioxidant and free radical scavenging activities, this partly deacetylated form of chitin is a prospective material for surface modification [112,113]. Chitosan can form functional coatings on Ti surface owing to the existence of amine groups in chitosan polymer chains, which are capable of developing covalent bonds with Ti via silanization [114]. Reasonable mechanisms for antioxidative action of chitosan include presence of intra-molecular hydrogen bonding [115], residual-free amino groups in water-soluble chitosan which may induce metal chelation [116] and the ability of NH_2_ amino groups to react with hydroxyl groups (OH^−^) to generate stable macromolecule radicals [117].

Lieder et al. studied the effect of degree of deacetylation of chitosan membranes coated on Ti surfaces which resulted in an improved fibronectin adsorption, cellular attachment and proliferation, but with no instigation of spontaneous osteogenic differentiation [114]. Chitosan coating (85–90% deacetylated) on porous Ti surface evidenced excellent antioxidant effect and favored osteoblast activity under diabetic conditions through reactivation of P13K/AKT pathway [118]. The study elucidated that chitosan can play a role in the reactivation of P13K/AKT pathway which mediates diabetes-induced ROS overproduction at bone-implant interface (Figure 3c,d). A multi-step layer-by-layer self-assembly was employed to deposit biofunctional composite coatings of chitosan and alginate enriched with caffeic acid on Ti-6Al-7Nb surface [119].

Multiple steps consisted of piranha solution treatment of Ti alloy surface, plasma chemical activation and dip coating. Antioxidant activity measured in terms of DPPH-scavenging activity was higher for chitosan coating due to its potent reducing activity by hydrogen-donating ability. Conjugation of caffeic acid on chitosan resulted in the generation of amide linkages, increasing the amount of electron-donating groups. Another chitosan-based composite coating consisting of chitosan-catechol, gelatin and hydroxyapatite nanofibres deposited on Ti substrates exhibited high level of ROS scavenging activity and decreased oxidative damage on cellular level as displayed in Figure 3a,b [51]. This coating was able to retain increased level of p-FAK (assists in cell spreading and migration) and p-Akt (control cell survival and apoptosis) compared to pure Ti. The developed multilayer coating improved cell matrix adhesion and intercellular adhesion, while attenuating ROS-induced damage by interfering expressions of integrin αv and β3, cadherin genes, anti-apoptotic and pro-apoptotic gene amounts. Electrophoretic deposition technique is also explored recently to coat chitosan-based composite coatings with hydroxyapatite, graphene and gentamycin [120].

#### 3.1.3. Proanthocyanidin

Proanthocyanidin is condensed tannins (comprising of flavan-3-ol monomeric units), which belongs to the class of naturally occurring polyphenol flavonoid (non-thiol natural antioxidant), is found abundantly in berries and fruits [121,122]. Proanthocyanidin possesses excellent ROS scavenging activity, can regulate macrophage behavior and is capable of stimulating bone formation under oxidative stress conditions [123,124,125,126]. Tang et al. reported layer-by-layer self-assembly method to deposit hyaluronic acid/chitosan multilayers with proanthocyanidins [127]. The three-dimensional multilayered network of hyaluronic acid/chitosan on Ti surface facilitated proanthocyanidin incorporation into the micro interspaces between hyaluronic acid and chitosan, eventually leading to its controlled release. Proanthocyanidin incorporation is based on the electrostatic interaction between reactive OH^−^ radical in proanthocyanidin and positive amine groups in chitosan. Layer-by-layer assembly is a self-assembly technique based on the electrostatic attractions (polyanions and polycations) between the assembled components to generate polyelectrolyte multilayers. Layer-by-layer technique involves charging Ti substrates by conjugating polyethylenimine for the purpose of obtaining higher binding forces. A sustained release of proanthocyanidin for a prolonged period of 14 days and mitigation of ROS-mediated inflammatory response were inferred. In other work, layer-by-layer technique was employed to integrate collagen type-II with proanthocyanidin which assisted in accelerating proliferation and osteogenic differentiation via Wnt/b-catenin signaling pathway and improved bone generation in vivo [128]. A novel covalent-conjugation strategy is reported to immobilize chitosan-encapsulated proanthocyanidin on Ti surface based on coupling agents (3-aminopropyl) triethoxysilane and glutaraldehyde [129]. Effective attenuation of the inhibitory effect of oxidative stress was induced by proanthocyanidin by the decrease of p53 gene expression. This study also indicated the improved stability of covalently immobilized coatings with improved wear and compression resistances attributed to strong chemical bonding and possessed the advantage of using nanoparticles as roller bearings.

## 4. Inorganic Materials for Surface Engineering Antioxidative Properties

### 4.1. Ceria Based Coatings

Cerium is a rare earth metallic element in lanthanide series and can exist in either free metal or metallic oxide form. Cerium possesses dual oxidation states: trivalent cerium sesquioxide (cerous Ce^3+^) and tetravalent cerium dioxide (ceric Ce^4+^) forms. Cerium oxide nanoparticles have received widespread attention for biocompatibility improvement, ophthalmic applications [130], cardiovascular pathology, treating neurodegenerative disorders and spinal cord injury owing to its ROS-scavenging ability [131,132]. The role of cerium oxide nanoparticles to effectuate ROS-scavenging activity and antioxidant mimicking role has been extensively reviewed by Nelson et al. [133]. Cerium oxide nanoparticles exhibit rapid and expedient switches in oxidation state between Ce^3+^ and Ce^4+^ during redox reactions. Owing to its lower reduction potential, cerium oxide exhibits redox-cycling property.

Cerium oxide is one of the most interesting oxides due to the presence of oxygen vacancy defects (which can be quickly generated and eliminated), and it can act as an oxygen buffer. The presence of oxygen vacancy sites on nanoceria lattice is responsible for the unusual catalytic activity of this class of material which is dependent on the efficient supply of lattice oxygen at reaction sites governed by the formation of oxygen vacancy sites [134]. Catalytic reaction of cerium oxide nanoparticles with super oxide anion (O^2−^) and hydrogen peroxide (H_2_O_2_) mimics biological action of SOD-mimetic and catalase thereby protecting cells against ROS induced damage [135]. Multi-enzymatic antioxidant activity is based on the ability of cerium oxide to rapidly switch between the multiple valence states. SOD mimic activity is elicited by a shift from Ce^3+^ to Ce^4+^ (scavenging of O^2−^), and catalase mimic activity is induced by a shift from Ce^4+^ to Ce^3+^ (deactivating hydrogen peroxide) [135,136,137]. SOD and catalase mimic activity of cerium oxide nanoparticles is particularly relevant under physiological pH condition (pH-7.5), rendering ROS-scavenging properties and inhibiting inflammatory mediator production.

Plasma-sprayed cerium oxide coating with a hierarchical topography was developed for antioxidant surfaces to preserve the intracellular antioxidant defense system [138]. Ceria oxide coating was found to be successful in decreasing SOD activity, reducing ROS generation and suppressing malondialdehyde development in hydrogen peroxide-treated osteoblasts. Li et al. reported magnetron sputtering (2, 3 and 5 min, ≈10^−^^4^ Pa) and vacuum annealing (450 °C) to deposit tiny homogenously distributed cerium oxide nanoparticle coatings with varying surface Ce^4+^/Ce^3+^ ratios by tuning of deposition time [131]. Quite interestingly, the Ce^4+^/Ce^3+^ ratio in this work reported the opposite trend for SOD and catalase mimetic activity. This work also highlighted the effective antioxidative mechanism of cerium oxide only when SOD and catalase mimetic activities are coordinated (H_2_O_2_ decomposition rate ≥ generation rate). The observed Ce^4+^/Ce^3+^ ratio resulted in improved cytocompatibility, new bone formation, bone integration and upregulation of osteogenic genes and protein expressions (Figure 4a). Apart from the surface chemistry, the shape of ceria-based nanoparticles has also been reported to influence its ROS scavenging activity. Nanowire-shaped ceria is reported to occupy the extracellular space as its cellular internalization rate is slow [139]. Hence, nanowire-shaped ceria present on the cell surface can level down the hydrogen peroxide molecules and induce ROS consumption as schematically shown in Figure 4b. Spin coating represents a quick and facile surface modification technique to obtain coatings of thickness ranging from a few nanometers to few micrometers. Spin coating technique was used to deposit hydrothermally synthesized nano-CeO_2_ with varying morphologies (nanorod, nano-cube and nano-octahedra) which yielded uniform coatings in Ti surfaces [140]. Total antioxidant capacity was in the order of nano-octahedron > nano-cube > nanorod. The anti-inflammation ability was correlated to the relative Ce^3+^ or Ce^4+^ content (XPS results displayed in Figure 4c) which, in turn, was influenced by the particle size and exposed crystalline lattice planes. With decreased particles size, Ce^3+^ content increased and rendered nano-octahedron improved SOD mimetic activity.

High energy Ce plasma was used to develop cerium-modified Ti surface by using plasma immersion ion implantation technology [141]. A shift in the appearance of surface from nano-grains to nano-pits was noted, with treatment time increased from 30 to 60 and 90 min. Cerium implantation on Ti surface resulted in reducing the hydroxyl radical generation on Ti surface with increase in plasma immersion time. Reduction in fluorescence intensities and enhanced protection of E.coli model from oxidative stress are attributed to the improved corrosion resistance of the modified surface and the capability of the CeO_x_ on Ti surface to consume hydroxyl radical and hydrogen peroxide. In a similar work, atmospheric plasma was used to deposit cerium oxide-incorporated calcium silicate coating on Ti-6Al-4V substrates [142]. The developed surfaces demonstrated good biochemical stability, cellular viability and antibacterial activity against *E. faecalis*. Recently metal organic framework (MOF) coating was developed in situ on Ti surface to develop bio-responsive Ce/Sr incorporated bio-MOF coating based on hydrothermal technique [143]. Hydrothermally treated Ce-MOF and Ce/Sr-MOF revealed a Ce^4+^/Ce^3+^ ratio of 1.186 and 2.76, respectively. Both of these Ce-containing surfaces revealed excellent H_2_O_2_ decomposition, superoxide anion disintegrating capacity, 80% of radical clearance during DPPH assay and persistence of antioxidant activity with TMB assay.

### 4.2. Silica

Various silicon based coatings have been explored for biocompatibility applications such as amorphous silicon oxygen thin films (a-SiOx) [144,145], calcium silicate [146], sol-gel-based silica bioglasses [147], silicon nitride [148], etc. Silicon is an important element which possesses an influential role in the activity of SOD to improve the ROS scavenging ability. Ilias et al. studied the plasma-enhanced chemical vapor deposition of amorphous silicon oxynitride in view of attaining rapid bone regeneration and fracture healing [149]. This study is the first of its kind to reveal the dependence of Si^4+^ on SOD1 to improve osteogenesis. For an effective bone healing, a sustained released of Si^4+^ should be ensured from the implant surface. Nitrogen incorporation into amorphous silica effectuated a continual Si^4+^ release which can be fine-tuned based on the surface chemistry (O/N ratio), and thickness of deposited film dictates the total release period. Plasma-enhanced chemical vapor deposition was similarly utilized by the same research group to develop coatings in the form of silicon oxynitrophosphide [150]. Up-regulation of SOD1 and cat-1 was observed in cells exposed to silicon oxynitrophosphide with varying oxygen and nitrogen contents. In other work, a radio frequency plasma-enhanced chemical vapor deposition (RF-PECVD) method which makes use of silane as Si source was used to deposit hydrogenated amorphous silicon coatings on Ti-6Al-4V substrate [151]. Hydrogen incorporation into the coating resulted in lower surface oxidation and amorphous silicon coating influenced fibroblasts with no significant effect on keratinocytes. A table enlisting the summary of advantages and limitations of different organic and inorganic coatings/materials described is provided in Table 2.

## 5. Conclusions and Future Perspectives

There are several titanium-surface modification techniques being used which can be classified as mechanical (polishing, blasting, peening), chemical (chemical treatment, sol-gel, anodic oxidation, chemical vapor deposition) and physical (thermal spraying, plasma spraying, physical vapor deposition, evaporation, ion plating, sputtering, glow discharge plasma, ion implantation and deposition) techniques [57,163]. In spite of the fact that several surface modification strategies are being researched with focus on antibacterial and cytocompatible surfaces, Ti surfaces with improved antioxidant properties require further research focus. The most common techniques explored on depositing functional molecules on the Ti surfaces are based on physical adsorption, based on weak hydrogen bonding and van der Waals forces. This is a limiting factor as it restricts the bond strength and coating life, which will potentially affect the efficacy of the implant. This can be tackled based on chemical immobilization via covalent bonding, in which case a more sustained release of functional molecules can be achieved as compared to physical adsorption techniques.

A critical limitation hampering the potentiality of Ti and its alloys is the inferior wear resistance to be used in articulating surfaces. In spite of the fact that various organic coatings on Ti are bioactive and can develop antioxidant activity, these coatings are mechanically instable, which is a particularly relevant aspect to be considered in terms of wear resistance. During surgical procedures, these implants often encounter mechanical forces of up to 15 N, which will critically affect the life of the coatings and sometimes can lead to coating spalling [129]. One way to tackle this will be the immobilization of such molecules on an already-modified surface layer [163]. Hence, a prospective idea is the development of bi-layer coating consisting of (a) an inner wear/corrosion resistant layer and (b) an outer bioactive layer with antioxidant activity. More research should be focused towards the extraction of exogenous antioxidants (mainly derived from food and medicinal plants, such as fruits, vegetables, cereals, mushrooms, beverages, flowers, spices and traditional medicinal herbs [164]) and its immobilization on Ti surface. Increasing the complexity of a surface modification process will render the process difficult and expensive for rapid commercialization.

One of the critical factors to be assessed while developing such surface is the effect of surface oxide layer on Ti surface. Ti and its alloys, when exposed to air, form a spontaneous native titanium dioxide (TiO_2_) layer with thickness in the range of 2–20 nm. This possesses a profound influence on the binding of molecules on the Ti surface and coating adhesion. Antioxidant release kinetics should also be given prior focus, as burst release in physiological environment can induce harmful enzymatic imbalances. More computational studies focused towards the stimulatory effect of various prospective coating materials on inducing oxidative stress and ROS need emphasis. Apart from these aspects, various factors to stimulate physiological conditions to assess the antioxidant activity of the developed surfaces shall be incorporated in studies, as synergetic influence of factors can alter the ROS scavenging activity.

Despite the beneficial aspects possessed by Ti and its alloy for load-bearing implant applications, there is a plenty of room for investigating the complex biological phenomena associated with ROS activity in the physiological environment. Most of the published works intended to improve the antioxidant properties of Ti surface are based on organic materials such as tannic acid, chitosan and proanthocyanidin and inorganic coatings based on ceria and silica. Future relevant research trends can be foreseen in improving the mechanical stability and controlled drug elution associated with organic coatings. It is also highly desirable that multifactorial aspects in a real physiological environment shall be considered while assessing the ROS scavenging activity of the developed surfaces. Overall, it is suggested to consider the ROS generation and antioxidant aspects with more research in this direction to develop an efficient implant surface of metallic biomaterials for improving the clinical efficiency. Most importantly, complex challenges associated with translation of lab research to clinical practice demands effective collaboration between material scientists, engineers, biologists and clinicians.

## Figures and Tables

**Figure 1 jfb-14-00344-f001:**
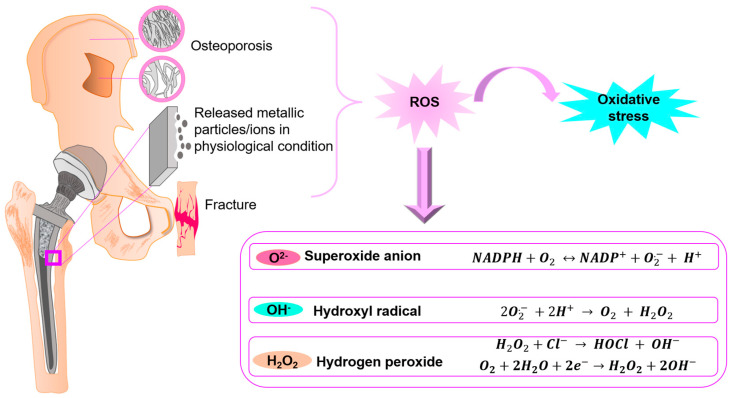
Schematic depicting the ROS generation associated with Ti implants with associated biochemical reactions.

**Figure 2 jfb-14-00344-f002:**
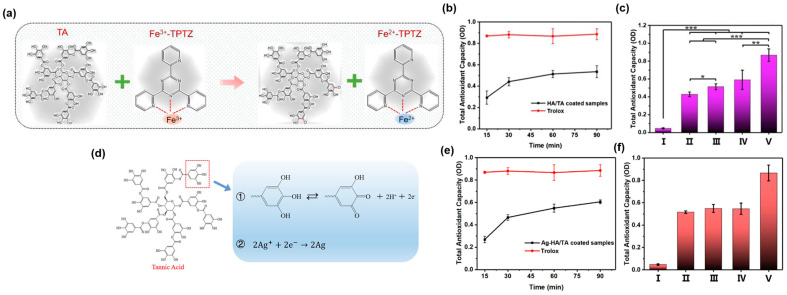
Antioxidant activity of HAP/tannic acid composite coating. (**a**) Mechanism illustrating antioxidant activity; (**b**,**c**) antioxidant activity as a function of incubation time of HAP/tannic acid coating compared to Trolox; (**c**) antioxidant activity of (I) negative control (deionized water), (II–IV) 10, 30 and 50 g/L of tannic acid, (V) positive control (Trolox). Data reported as means ± standard deviations, n = 3 (* *p* < 0.05, ** *p* < 0.01, *** *p* < 0.001). Reprinted from [101] with permission from Elsevier. (**d**) Formation mechanism depiction of Ag nanoparticles; (**e**) antioxidant activity with respect to incubation time of Ag/HAP/tannic acid coating compared to Trolox; and (**f**) antioxidant activity of (I) negative control (deionized water), (II–IV) 0.05, 0.1 and 0.2 M AgNO_3_, (V) positive control (Trolox). Reprinted from [53] with permission from Elsevier.

**Figure 3 jfb-14-00344-f003:**
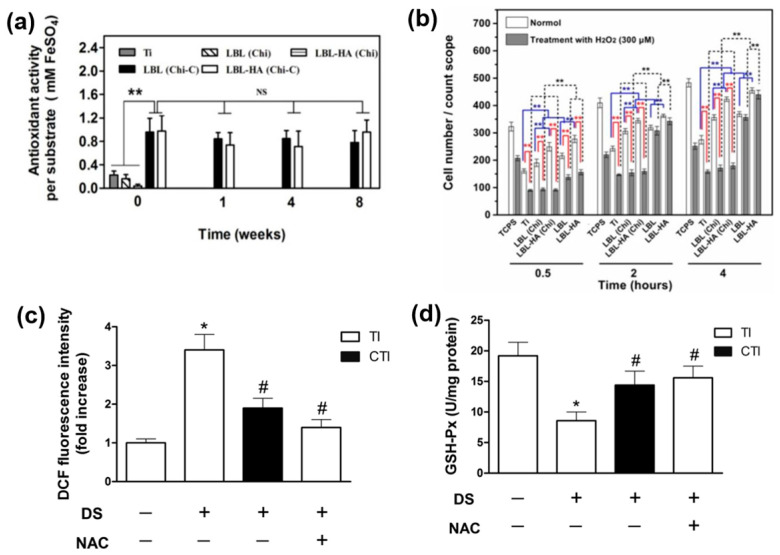
Antioxidant activity of chitosan based surfaces on Ti. (**a**) Antioxidant activity measured over a period of 8 weeks; (**b**) osteoblast interaction with and without treatment of hydrogen peroxide. Reprinted from [51] with permission from Elsevier. (**c**) intracellular ROS activity measured by DCF fluorescence intensity and (**d**) intracellular GSH-Px activity to quantify the rate of oxidation of the reduced glutathione to the oxidized glutathione by H_2_O_2_ (* *p* < 0.05 vs. TI + NS; # *p* < 0.05 vs. TI + DS; ** *p* < 0.01). Reprinted from [118] with permission from Elsevier.

**Figure 4 jfb-14-00344-f004:**
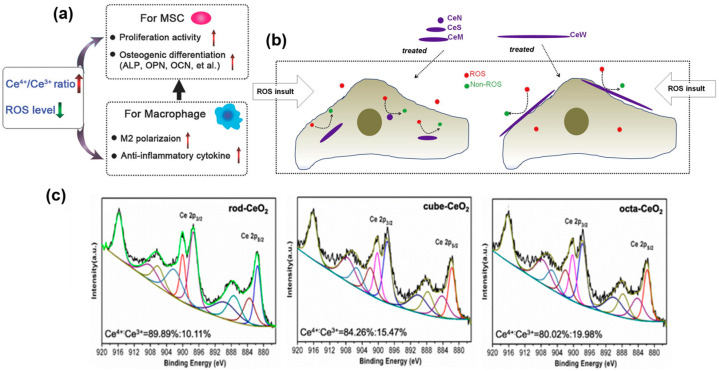
Influence of ceria surface chemistry and shape on the ROS scavenging activity. (**a**) Influence of Ce^4+^/Ce^3+^ ratio on MSC and macrophage. Reprinted from [131] with permission from John Wiley & Sons. (**b**) X-ray-photoelectron spectroscopy analysis of nanorod, nano-cube and nano-octahedron-shaped ceria with varying Ce^4+^/Ce^3+^ ratio. Reprinted from [139] with permission from Elsevier. (**c**) Schematic depicting the scavenging of extracellular matrix ROS by nanowire-shaped ceria present at the cell surface. Reprinted from [140] with permission from Elsevier.

**Table 1 jfb-14-00344-t001:** Various reported techniques used for developing antioxidant surfaces on Ti/Ti alloy surfaces.

Technique	Description	Ref.
Layer-by-layer technique	Bottom-up adsorption technique which involves the development of multi-layered (layers of oppositely charged species) thin films bound together through electrostatic interactions.	[79,80]
Immersion/dip coating	Solution-based deposition method which involves the immersion of substrate into a solution of material to be coated which depends on parameters such as dwelling time, substrate-withdrawal speed, number of dip-coating cycles and coating evaporation factor.	[81,82]
Spin coating	A technique which uses centrifugal force for deposition, in which a suspension is dropped from top into the rotating substrate, and the resulting centrifugal force will assist in spreading out the coating on the substrate, thereby coating it. The process is dependent on parameters such as dispense volume, spin speed, solution viscosity, solution concentration, spin time, etc.	[83,84,85]
Plasma immersion ion implantation (PIII)	Method to improve biocompatibility aspects of material surfaces by immersing in a plasma environment and applying negative-high-voltage pulsed bias. Compared to traditional plasma techniques, PIII can extend to some tens of nanometers beneath sample surface and can treat complex geometries.	[86,87,88]
Plasma enhanced chemical vapor deposition	Low temperature chemical vapor deposition in which plasma is used to drive chemical reactions between plasma-generated-reactive species and substrate instead of high temperatures.	[89,90,91]

**Table 2 jfb-14-00344-t002:** Summary of various coatings used for antioxidant properties.

Type of Coating/Surface	Antioxidant Mechanism	Advantages	Limitations	Ref.
Tannic acid	Ability to chelate metal ions such as Fe(II), thereby interfering with one of the reaction steps in the Fenton reaction and thereby slowing oxidation	Antibacterial, antioxidant, high hemostatic efficiency, anticancer property, regenerative potential	Weak lipid solubility, low bioavailability, and short half-life, release rate should be controlled to exclude potential cytotoxicity, unstable adhesion	[152,153,154,155]
Chitosan	Reasonable mechanisms include presence of intra-molecular hydrogen bonding, metal chelation, ability of NH_2_ amino groups to react with hydroxyl groups	Biological activity, antimicrobial activity, hydrophilicity, and biodegradability	Delamination, unstable adhesion	[156,157,158]
Proanthocyanidin	By scavenging free radicals and by modifying signaling pathways, including those involving nuclear factor erythroid 2-related factor 2 (Nrf2), mitogen-activated protein kinase (MAPK), nuclear factor-kappaB (NF-κB), and phosphoinositide 3-kinase (PI3K)/Akt	Antioxidant, anticancer, antidiabetic, neuroprotective, and antimicrobial	High cost, low chemical stability and limited binding sites, difficulties in resolving the chemical labeling pattern of PAs with their proposed biosynthetic pathway, and defining the subcellular sites of biosynthesis	[121,124,159]
Ceria	Ability to rapidly switch between multiple valence states. SOD mimic activity is elicited by a shift from Ce^3+^ to Ce^4+^ (scavenging of O^2−^) and catalase mimic activity is induced by a shift from Ce^4+^ to Ce^3+^ (deactivating hydrogen peroxide)	Antioxidant, anticancer and anti-inflammatory properties, biosensors	Toxicity associated with small-sized nano-ceria	[160,161]
Silica	Hydroxylation degree, By regulation of antioxidants enzyme activity	Accelerated bone fracture healing, biomineral synthesis	Lipid peroxidation induced toxicity	[149,162]

## Data Availability

The data presented in this study are available on request from the corresponding author.

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
