# Peer review of "Engineering Antioxidant Surfaces for Titanium-Based Metallic Biomaterials"

_jfb, 2023, doi:10.3390/jfb14070344_

Round 1

Reviewer 1 Report

jfb-2458562

In this manuscript, the authors provide a comprehensive review regarding the " Engineering antioxidant surfaces for titanium-based metallic 2 biomaterials ". It is eye-catching and exciting to a researcher in this field. However, some corrections and explanations must be done previous to publish the paper in the J. Funct. Biomater. Some of my comments and questions on this manuscript are as follows:

1.    Authors should identify gaps in existing studies for potential future research. For example, in conclusion, the recommendation for future research is quite general.

2.    The authors did not explain the novelty and significance of their work in the introduction section. Moreover, this section is not cohesive. Indeed, this section is intended to "convey the core findings of the paper," i.e., reflect the best novelty of this review paper in a concise form. The authors shall show the work's best novelty, such as how your research. advances the state-of-the-art of the topic/area and /or how much better is your work compared with peer researchers on the same or similar topics. At the end of this section, the main objective of this study must be mentioned.

3.   I suggest that the author present the further properties and applications of Ti-based implant materials with antioxidant activity in various fields including biomedical applications. More in-depth discussion of related previous works in this regard is required.

4.  Well-organized discussion should be added regarding the ROS generation aspects associated with Ti-based metallic biomaterials, combined with recent works and scientific progress in the area.

5. I suggest that the author present the schematic illustration regarding antioxidant activity of composite/coated based surfaces on Ti-based implant materials.

6. The author could provide more information about the various organic and inorganic coatings/ materials for effective antioxidant surfaces for Ti-based implant materials. The author could provide a summary of advantages and limitations of different organic and inorganic coatings/ materials for effective antioxidant surfaces for Ti-based implant materials. 

7. The manuscript contains huge textual documentation, which makes it very boring for the readers. I would suggest the authors add some comprehensive tables. The content of the present table is not sufficient regarding the various organic and inorganic coatings/ materials for effective antioxidant surfaces for Ti-based implant materials.

8. Some references about antioxidants and antimicrobial based materials may be useful for this review article: Antioxidants 2020; 9 (12), 1309; Advanced Engineering Materials 2022, 24 (6), 2101460. In addition, surprisingly small reference to the J. Funct. Biomater. (JFB) in the literature despite the large relevant literature there. This should be improved. There are several important papers in recent literature.

Author Response

We would like to thank the editor and the reviewers for their insightful comments and suggestions for the improvement of the manuscript. We have addressed the queries raised by the reviewers point by point and manuscript has been modified accordingly. The changes are marked up using the *Track Changes* function.

Reviewer 1:

In this manuscript, the authors provide a comprehensive review regarding the " Engineering antioxidant surfaces for titanium-based metallic 2 biomaterials ". It is eye-catching and exciting to a researcher in this field. However, some corrections and explanations must be done previous to publish the paper in the J. Funct. Biomater. Some of my comments and questions on this manuscript are as follows:

  1. Authors should identify gaps in existing studies for potential future research. For example, in conclusion, the recommendation for future research is quite general.

Response: We would like to thank the reviewer for pointing out this aspect. As suggested by the reviewer, the revised review manuscript is incorporated with prospects for future research. The conclusion section has been modified by addition of more details.

  1. The authors did not explain the novelty and significance of their work in the introduction section. Moreover, this section is not cohesive. Indeed, this section is intended to "convey the core findings of the paper," i.e., reflect the best novelty of this review paper in a concise form. The authors shall show the work's best novelty, such as how your research. advances the state-of-the-art of the topic/area and /or how much better is your work compared with peer researchers on the same or similar topics. At the end of this section, the main objective of this study must be mentioned.

Response: We would like to convey here that the novelty aspect of the present review paper is provided in the last paragraph of the Introduction section. As recommended by the reviewer we have modified this section, by highlighting how the present review advance the state-of-the-art Ti surface modification. Finally at the end of introduction, objective has been updated. 

  1. I suggest that the author present the further properties and applications of Ti-based implant materials with antioxidant activity in various fields including biomedical applications. More in-depth discussion of related previous works in this regard is required.

Response: As recommended by the reviewer, the application of Ti and its alloys for biomedical applications has been elaborated.

  1. Well-organized discussion should be added regarding the ROS generation aspects associated with Ti-based metallic biomaterials, combined with recent works and scientific progress in the area.

Response: The section describing ROS generation aspects associated with Ti-based metallic biomaterials has been elaborated by citing more relevant research works.

  1. I suggest that the author present the schematic illustration regarding antioxidant activity of composite/coated based surfaces on Ti-based implant materials.

Response: As recommended by the reviewer, a schematic illustration has been provided as Figure 1, depicting the ROS generation and resulting antioxidant activity of Ti-based implant materials. A concise description of antioxidant activity of various surfaces is summarised in Table 2.

  1. The author could provide more information about the various organic and inorganic coatings/ materials for effective antioxidant surfaces for Ti-based implant materials. The author could provide a summary of advantages and limitations of different organic and inorganic coatings/ materials for effective antioxidant surfaces for Ti-based implant materials. 

Response: We would like to convey here that, there are only a few coatings/surfaces specifically developed for improving the antioxidant aspects of Ti surfaces. As per the Reviewer’s suggestion a table enlisting the advantages and limitations of different coatings along with the antioxidant mechanism have been incorporated as shown in Table 2 in the revised manuscript. In addition to this, various surface modification strategies utilised for developing antioxidant surfaces has been updated in Table 1.

  1. The manuscript contains huge textual documentation, which makes it very boring for the readers. I would suggest the authors add some comprehensive tables. The content of the present table is not sufficient regarding the various organic and inorganic coatings/ materials for effective antioxidant surfaces for Ti-based implant materials.

Response: Tables enlisting various surface modification strategies and materials for antioxidant surfaces have been incorporated in the revised version of the review paper.

  1. Some references about antioxidants and antimicrobial based materials may be useful for this review article: Antioxidants 2020; 9 (12), 1309; Advanced Engineering Materials 2022, 24 (6), 2101460. In addition, surprisingly small reference to the J. Funct. Biomater. (JFB) in the literature despite the large relevant literature there. This should be improved. There are several important papers in recent literature.

Response: As recommended by the reviewer, the references have been updated.

Reviewer 2 Report

Reactive oxygen species (ROS) pose significant concerns in relation to the failure of orthopedic metallic implants. Effective removal of ROS is a critical step in the performance of these implants. The review article by Vishnu et al. offers a concise summary primarily focusing on ROS associated with titanium implants, encompassing their generation and various organic and inorganic coating methods to enhance the antioxidant properties of titanium surfaces. While the article provides valuable information for readers and may be beneficial to researchers and students in this field, it is currently limited in scope. To enhance its value as a review article, the authors should address the following concerns:

  1. Surface coating as a method to enhance implant antioxidation performance should be elaborated upon, including the rationale behind this approach, overall progress in the field, and a comparison with other methods. A comprehensive summary and critical evaluation of this technique, including its advantages and disadvantages, should be provided.
  2. The review should include an overview of the biomedical applications of titanium implants, highlighting their usage and significance in various contexts.
  3. It is recommended to include a figure illustrating the generation and metabolism of ROS related to titanium implants. Additionally, the importance of hydroxyl radicals and singlet oxygen in the context of titanium implants should be addressed.
  4. The mechanism by which coating materials clear ROS is a particularly interesting aspect. The authors should provide a clear description of the mechanism, preferably accompanied by schematic representations to aid understanding.
  5. A section detailing the various coating strategies should be included, outlining different approaches and their specific characteristics.
  6. While discussing coating materials, it is important to consider the wide range of available options. Therefore, the authors should explain why they have chosen to focus on only three types of organic materials and two types of inorganic materials in their review.
  7. The conclusion and perspective sections could be combined, allowing for a more comprehensive discussion. The perspective section should be expanded to critically address remaining issues in the field, propose potential solutions, and outline future research directions.
  8. The quality of the figures in the manuscript is currently inadequate, with some images being blurred and containing noticeable artifacts. The authors are advised to redraw the figures to improve their quality. Additionally, instead of directly copying figures from journals, the authors should consider creating their own illustrations for structures and profiles where applicable.

By adequately addressing these concerns, the authors can significantly improve the manuscript and ensure its suitability for publication.

Author Response

We would like to thank the editor and the reviewers for their insightful comments and suggestions for the improvement of the manuscript. We have addressed the queries raised by the reviewers point by point and manuscript has been modified accordingly. The changes are marked up using the *Track Changes* function.

Reviewer 2:

Reactive oxygen species (ROS) pose significant concerns in relation to the failure of orthopedic metallic implants. Effective removal of ROS is a critical step in the performance of these implants. The review article by Vishnu et al. offers a concise summary primarily focusing on ROS associated with titanium implants, encompassing their generation and various organic and inorganic coating methods to enhance the antioxidant properties of titanium surfaces. While the article provides valuable information for readers and may be beneficial to researchers and students in this field, it is currently limited in scope. To enhance its value as a review article, the authors should address the following concerns:

  1. Surface coating as a method to enhance implant antioxidation performance should be elaborated upon, including the rationale behind this approach, overall progress in the field, and a comparison with other methods. A comprehensive summary and critical evaluation of this technique, including its advantages and disadvantages, should be provided.

Response: We would like to thank the reviewer for pointing out these missing aspects. Rationale behind surface modification approach to improve the antioxidation performance has been included in the final paragraph of Introduction section. A table enlisting the advantages and limitations of various coatings have been summarized in the revised manuscript.

  1. The review should include an overview of the biomedical applications of titanium implants, highlighting their usage and significance in various contexts.

Response: The biomedical applications of various titanium-based alloys have been elaborated in Section 2, in the revised manuscript as recommended by the reviewer.

  1. It is recommended to include a figure illustrating the generation and metabolism of ROS related to titanium implants. Additionally, the importance of hydroxyl radicals and singlet oxygen in the context of titanium implants should be addressed.

Response: We would like to thank the reviewer for pointing out this. A schematic illustration of ROS generation and metabolism associated with Ti implant surfaces has been included in revised manuscript.

  1. The mechanism by which coating materials clear ROS is a particularly interesting aspect. The authors should provide a clear description of the mechanism, preferably accompanied by schematic representations to aid understanding.

Response: ROS scavenging mechanism is a complex mechanism which varies with the distinctive type of surface used. As recommended by the reviewer, ROS scavenging mechanism of various coating material/surface has been updated in Table 2.

  1. A section detailing the various coating strategies should be included, outlining different approaches and their specific characteristics.

Response: As suggested by the Reviewer, an introductory section outlining various approaches and specific characteristics have been included in the revised manuscript. A table has been added (Table 1) to detail the various surface modification strategies utilised for generating antioxidant surfaces.

  1. While discussing coating materials, it is important to consider the wide range of available options. Therefore, the authors should explain why they have chosen to focus on only three types of organic materials and two types of inorganic materials in their review.

Response: We would like to convey here that it has already been mentioned at the last part of introduction section that this review is specifically focused on surface modification strategies specifically developed for antioxidant surfaces. There is not a vast diversity of materials explored for these applications on metallic implant surfaces and hence the major organic (tannic acid, chitosan, proanthocyanidin) and inorganic (ceria, silica) coating materials have been explained in the current review.  

  1. The conclusion and perspective sections could be combined, allowing for a more comprehensive discussion. The perspective section should be expanded to critically address remaining issues in the field, propose potential solutions, and outline future research directions.

Response: As recommended by the reviewer, the conclusion and perspective sections have been combined and the section is expanded by addressing the points mentioned.

  1. The quality of the figures in the manuscript is currently inadequate, with some images being blurred and containing noticeable artifacts. The authors are advised to redraw the figures to improve their quality. Additionally, instead of directly copying figures from journals, the authors should consider creating their own illustrations for structures and profiles where applicable.

Response: High resolution images have been incorporated as suggested.

By adequately addressing these concerns, the authors can significantly improve the manuscript and ensure its suitability for publication.

Round 2

Reviewer 2 Report

The revised manuscript is recommended for publication.